# Neutrophils in Dendritic Cell-Based Cancer Vaccination: The Potential Roles of Neutrophil Extracellular Trap Formation

**DOI:** 10.3390/ijms24020896

**Published:** 2023-01-04

**Authors:** Lily Chan, Geoffrey A. Wood, Sarah K. Wootton, Byram W. Bridle, Khalil Karimi

**Affiliations:** 1Department of Pathobiology, Ontario Veterinary College, University of Guelph, Guelph, ON N1G 2W1, Canada; 2ImmunoCeutica Inc., Cambridge, ON N1T 1N6, Canada

**Keywords:** neutrophils, NET, dendritic cell vaccination, immunotherapy, plasticity

## Abstract

Neutrophils have conflicting roles in the context of cancers, where they have been associated with contributing to both anti-tumor and pro-tumor responses. Their functional heterogenicity is plastic and can be manipulated by environmental stimuli, which has fueled an area of research investigating therapeutic strategies targeting neutrophils. Dendritic cell (DC)-based cancer vaccination is an immunotherapy that has exhibited clinical promise but has shown limited clinical efficacy. Enhancing our understanding of the communications occurring during DC cancer vaccination can uncover opportunities for enhancing the DC vaccine platform. There have been observed communications between neutrophils and DCs during natural immune responses. However, their crosstalk has been poorly studied in the context of DC vaccination. Here, we review the dual functionality of neutrophils in the context of cancers, describe the crosstalk between neutrophils and DCs during immune responses, and discuss their implications in DC cancer vaccination. This discussion will focus on how neutrophil extracellular traps can influence immune responses in the tumor microenvironment and what roles they may play in promoting or hindering DC vaccine-induced anti-tumor efficacy.

## 1. The Crosstalk of Neutrophils and Dendritic Cells during Immune Responses

Dendritic cells (DCs) are innate leukocytes critical in initiating immune responses directed toward tumor cells [1]. DCs are important in forming immune responses due to their superior antigen-presenting capacities. They are heterogeneous and can be divided into subsets; two main subdivisions are plasmacytoid DCs (pDCs) and conventional/classical DCs (cDCs). cDCs can be further divided into cDC1s and cDC2s. The other subset arises in monocyte-induced inflammatory conditions and is called monocyte-derived DCs (mo-DCs).

pDCs are significant producers of type I IFNs that upregulate antiviral mechanisms in cells. Therefore, pDCs are commonly associated with defense against viral infections. cDCs are mainly known for their ability to present antigens to T cells. cDC2s present exogenously acquired antigens via their major histocompatibility complex (MHC) class II. cDC1s are known for their ability to perform cross-presentation, which is a process that involves presenting exogenously acquired antigens on MHC class I and priming antigen-specific cytotoxic T lymphocyte (CTL) responses. Mo-DCs have been observed to have similarities to cDCs and can perform cross-presentation [2].

Neutrophils are an essential component of the innate immune system and have many receptors to recognize various pathogen- and damage-associated molecular patterns. Once at a site of inflammation, they become activated and release chemokines and other factors that will recruit leukocytes to the local area, including T cells, monocytes, macrophages, and other neutrophils [3]. They are phagocytic granulocytes abundant in human circulation and are known as immediate responders at sites of injury and inflammation. They have roles in recruiting other leukocytes, killing pathogens, and clearance of pathogens and debris. Neutrophils can release neutrophil extracellular traps (NETs) when activated. These NETs are composed of various components, including proteins from neutrophil granules, DNA, and neutrophil elastase. Production of NETs is a critical function of neutrophils in the defense against extracellular pathogens, as well as viruses during the extracellular phase of their life cycle [4]. Neutrophils usually have a short lifespan of approximately one day. However, inflammatory by-products or cytokines can activate them, and depending on the situation, their lifespan can be extended for a few days [5,6,7], allowing them to participate in more complex activities, for instance, in the multi-cellular crosstalk that occurs in tumor microenvironments (TMEs) [8,9]. In humans and mice, the role for neutrophils as critical regulators in the maturation and function of natural killer (NK) cells has been demonstrated in steady state [10]. Murine [11] and human [12] neutrophils have also been observed to have a capacity for antigen presentation.

Furthermore, in mouse models, neutrophils have been demonstrated to migrate to lymph nodes, promote CD4+ T cell responses, and regulate CD4+ T cell proliferation [13]. Compared to other leukocytes like monocytes, neutrophils are relatively low cytokine producers on a per-cell basis, but in instances like injury, they are abundant at sites of inflammation and are responsible for a large portion of the cytokines present due to the sheer number of neutrophils at the site [14,15]. Therefore, due to the timing and magnitude (in terms of cell number) of their responses, neutrophils could play important roles in setting the immunological stage for ensuing responses and manipulating an immunological environment through large amounts of cytokines. Thus, understanding the communications and immunobiology of neutrophil responses could have important implications for health and disease.

Although DCs are generally tissue-resident cells and neutrophils are a primary component of the blood, these leukocytes cross paths at sites of inflammation and in lymphoid organs [16,17]. Neutrophils will rapidly traffic to an area of injury or inflammation, where they have been demonstrated to have roles in the recruitment, activation, and maturation of DCs [3,16,17,18]. Neutrophils produce chemokines and anti-microbial products that recruit immature DCs to the local area, and immature DCs produce factors that also recruit neutrophils, which can amplify co-localization and promotion of crosstalk between neutrophils and DCs [16]. Observations that neutrophils can notify DCs that infection has occurred and promote DCs to stimulate T helper 1 (Th1) responses suggest that neutrophils can be contributing players in the organization of adaptive immune responses [18,19]. Neutrophils have been demonstrated to activate the maturation of mo-DCs in vitro in a process that involved tumor necrosis factor (TNF)-α and cell-to-cell contact through DC-specific intercellular adhesion molecule-3-grabbing non-integrin (SIGN)/CD209 and Mac-1. Moreover, these activated DCs were shown to support the formation of Th1 responses [20]. Therefore, this is a mechanism whereby neutrophils can impact adaptive immune responses through communication with DCs.

Studies of the interactions between neutrophils and mo-DCs in vitro have shown that neutrophils can stimulate DCs to upregulate costimulatory molecules such as CD40 and CD86 and promote their activation. Moreover, apoptotic neutrophils served as a source of antigens for DCs to capture and present [17]. This further supports the roles neutrophils can have in the activation of DCs and the formation of adaptive immunity. Although neutrophils have been observed to have direct functions in adaptive immunity, including recruitment and activation of Th1 cells, many of their functions in adaptive immunity rely on their interplay with DCs [19,21]. Intriguingly, Yang et al. observed that depletion of neutrophils increased antigen presentation by DCs, enhanced induction of CD4+ T cell responses, and increased the number and duration of DC and T cell contact interactions following injection of protein antigens in complete or incomplete Freud’s adjuvant in murine models [22]. These observations suggest that neutrophils can significantly influence the education of T cells by DCs, both negatively and positively. Therefore, the communications between neutrophils and DCs must be evaluated thoroughly in health and disease to properly assess their contributions and how their crosstalk could be targeted in therapeutic contexts.

## 2. Neutrophil Functional Heterogenicity in Cancers

Neutrophils are implicated in the pathogenesis of several diseases, such as rheumatic disease [23], coronavirus disease 2019 [24], systemic lupus erythematosus [25], and malaria [26]. Neutrophils have been shown to be a heterogeneous population of innate leukocytes. Their phenotypes and functions can be influenced by external cues in their microenvironments. This makes their roles in disease complex and signifies a need to understand their heterogenicity and plasticity to develop effective immunotherapies. In the context of cancers, neutrophils have progressively been established in the scientific literature as important players. A higher neutrophil to lymphocyte ratio has been observed to be associated with a poorer prognosis in various cancers [27,28,29] suggesting neutrophils are supportive of cancer progression. However, evidence demonstrates that neutrophils appear to have contradictory roles in cancers and tumor-associated neutrophils (TANs) have been observed to exist in two different functional states in TMEs that have been defined as two subsets of neutrophils, N1 and N2 [21,30,31,32]. Discriminating between N1 and N2 phenotypes based on surface markers is limited as specific profiles for the phenotypes have not been defined and widely accepted and the plasticity between the neutrophil subsets is extensive. Typically TANs are distinguished based on functional characteristics such as cytokine production [33]. N1s are associated with type 1 responses and are anti-tumorigenic and N2s are associated with type 2 responses and are pro-tumorigenic [21,30,31]. N2s can promote tumor growth and metastasis through the secretion of several factors and enzymes including matrix metalloproteinases, that promote angiogenesis and extracellular matrix remodeling [34,35]. Furthermore, neutrophils have been observed to promote evasion of tumors from the immune system by suppressing the anti-tumor functions of NK cells [35].

Neutrophil production of NETs is a critical process in microbial defense and important in other aspects of immunity. However, NETs have also been associated with some of the harmful roles neutrophils can play in the pathogenesis of different diseases [36,37,38]. Concerning cancers, NETs have been associated with cancer progression [39,40,41]. Using both human and murine-derived cells, in vitro experiments showed that neutrophils produced NETs, which appeared to block NK cells and CTLs from accessing the tumor cells and killing them. This was supported when transferred in vivo in murine models where tumors were observed to promote neutrophils in the TME to produce NETs to protect cancer cells from anti-tumor responses [40].

NETs also appear to promote angiogenesis [42,43] and can support cancer metastasis [39,44,45]. NETs have been implicated in supporting cancer metastases and recurrence through various mechanisms, including capture and sequestration of circulating cancerous cells [46] and promotion of dormant cancer cell awakening [39]. In vitro experiments using a human pancreatic carcinoma cell line showed that the cancerous cells induced neutrophil NET formation, and this was observed to promote migration of the cancer cells, angiogenesis, and thrombosis [47]. There has also been evidence that NETs interfere with immune checkpoint blockade treatment. In murine models of pancreatic ductal adenocarcinoma, interleukin (IL)-17 was demonstrated to promote neutrophil recruitment and NET formation in the TME and hinder CD8+ T cell responses. When IL-17/IL-17 receptor A was blocked, there were improved responses to immune checkpoint blockade therapy, indicating that neutrophils and NET production suppresses anti-tumor responses and combining immunotherapies with treatments targeting neutrophils/NETs could be beneficial. To investigate the translation of these findings, the serum of human patients with pancreatic ductal adenocarcinoma was examined ex vivo for stimulation of NETs and had significantly higher stimulation of NET formation and decreased NET degradation compared to healthy human controls, which suggests there is altered NET formation responses in patients with pancreatic ductal adenocarcinomas [48]. There have also been observations of increased NETs found in lung tissue and blood of patients with lung cancers compared to cancer-free patients. NETs and lung cancers were also examined in mouse models, and a crosstalk between the tumor cells and neutrophils that promoted NET formation was detected [49]. Therefore, it appears that cancerous cells can promote the formation of NETs, which promotes pro-tumorigenic responses and indicates a need to target NETs as an additional component to current therapies. However, further studies investigating the combined treatment with immunotherapy and an inhibitor of NET formation should be conducted to investigate the translatability to humans. NETs appear to have the capacity to hinder cytotoxic immune responses against tumor cells and promote the tumor’s survival, suggesting they can support the metastasis of tumor cells. Therefore, NET production may be a functional characteristic of N2s.

Conversely, neutrophils have also been associated with anti-tumor responses. N1s can produce proinflammatory cytokines and chemokines that induce recruitment and activation of many different leukocytes, including DCs, NK cells, CTLs, and Th1 cells that promote anti-tumor responses [50]. Neutrophils can also produce different factors, including reactive oxygen species that stimulate tumor cell death [21]. For instance, using murine models, N1s were observed to induce tumor cell apoptosis in vitro and reduce tumor growth in vivo, whereas N2s were shown to augment tumor growth in vivo [32]. In mouse models of sarcomas, neutrophils were shown to have important roles in supporting type I immune responses in the TME and promoting anti-tumor immunity. It was also demonstrated that neutrophils increased IL-12 production by macrophages and that this neutrophil-macrophage interaction promoted the type I polarization of unconventional αβT cells. This communication between neutrophils, macrophages, and unconventional αβT cells supported tumor resistance in this mouse model. These observations of neutrophils supporting type I anti-tumor immunity appeared to extend to some human tumor samples that were analyzed via RNA sequencing and from patient-derived cancer data sets [51]. Furthermore, TANs isolated from patients with early stage lung cancers could improve T cell responses by increasing the proliferation and activation of T cells and their production of IFNγ in in vitro experiments [9].

Neutrophils have also been described in terms of their relative density, where there are low-density neutrophils and high-density neutrophils that can be physically separated by centrifugation on a density gradient. Low-density neutrophils have been associated with multiple inflammatory diseases [25,52,53]. In the context of cancers, low-density neutrophils are associated with pro-tumorigenic responses and have immunosuppressive properties, whereas high-density neutrophils are associated with anti-tumor responses [54,55]. A research group profiled circulating neutrophils in patients with advanced lung cancers and evaluated high- and low-density neutrophils. Higher numbers of low-density neutrophils and higher neutrophil-to-leukocyte ratios were associated with advanced lung cancers and a worse prognosis. They also noted extensive plasticity between the high- and low-density neutrophils with phenotypic variability of cells between patients [56]. In murine lung cancer models, a switch between neutrophil densities as cancer progressed was observed, where high-density neutrophils became low-density neutrophils [54]. Furthermore, murine models demonstrated that low-density neutrophils have metastasis-promoting capacities [55].

Interestingly, external stimuli that neutrophils receive can influence their polarization to a N1 or N2 phenotype [57]. The TMEs take advantage of and will produce cytokines, such as tumor growth factor-β, that promote N1s to change to an N2 phenotype [50,58]. In the blood of patients with melanomas, type I IFN therapy appeared to promote a N1 phenotype in neutrophils and in vivo in murine tumor-bearing models it was demonstrated that type I IFNs influenced the polarity of neutrophils. There was also evidence that IFN-β could promote an anti-tumor phenotype in neutrophils [59]. Using primary human neutrophils and polarization cocktails, Ohms et al. [33] were able to polarize neutrophils in vitro to acquire either an N1 or N2 phenotype. The plasticity between N1 and N2 suggests the potential to therapeutically manipulate the phenotype of neutrophils to produce desired responses in the TME [60]. Overall, neutrophils play important roles in the TME and exist in increased numbers in patients with cancers. As a result, different therapeutic approaches that target neutrophils are being studied to enhance anti-tumor responses [58,60].

## 3. DC Vaccines

The DC subsets exist in the TME at varied concentrations, playing diverse roles, and have been observed to have various effects [61]. In the TME, cDC1s educate CTLs and produce cytokines that promote Th1 polarization, cDC2s function to present antigens to CD4^+^ T cells, pDCs have been observed to secrete type I IFN and promote Th1 responses, and mo-DCs generate proinflammatory cytokines and educate CTLs [62]. cDC1s are regarded as key players in anti-tumor immunity due to their ability to prime and activate antigen-specific cytotoxic T cell responses. In mouse models, cDC1s have not only been observed to be the antigen-presenting cell responsible for the education of tumor-specific CTLs, but they also have critical roles in the recruitment of CTLs to tumors [63]. The TME creates an immunosuppressive milieu that inhibits the functions of inflammatory DCs to prevent activation and education of leukocytes that would otherwise target tumor cells [64]. The TME also promotes an immunosuppressive phenotype of DCs that will induce immunological tolerance to tumor cells [1]. Furthermore, the TME will downregulate the production of chemotactic factors such as chemokine ligand (CCL)-4, which recruits DCs, as another mechanism to avoid the education of adaptive leukocytes that will target tumor cells [65].

The TME works to hinder immune responses against tumor cells through several mechanisms, a major one being the inhibition of the ability of DCs to activate innate leukocytes and educate CTLs to target tumor cells [66]. Clinical investigations into the potential of using DC-based vaccines as cancer immunotherapy began in the 1990s [67]. The general concept of DC-based cancer vaccines is to reactivate an immune system that has been suppressed by the TME and promote tumor-specific CTL responses [68]. DCs are of particular interest in immunotherapy because they are the most potent antigen-presenting cells and can present antigens through both MHC class I and II. They have the ability to cross-present antigens and can therefore present tumor-associated antigens they have obtained from the extracellular environment to CD8+ T cells and produce antigen-specific CTLs [68]. To generate DC-based vaccines, autologous DCs or DC precursors are isolated from a patient’s blood and DCs are cultured ex vivo. These DCs are matured, loaded with the desired antigens, and then re-administered to the patient. The DCs will travel through the lymphatics and prime antigen-specific responses directed towards tumor cells [69]. DC targeting vaccines provide a framework for improving vaccine responses against cancers and infectious diseases. DC vaccines can be loaded with cell lysate, mRNA, specific or associated antigens, and DCs transfected with vectors that have or are encoded with immunostimulatory factors or antigens. The most used DC vaccine platform involves the stimulation of ex vivo DCs with cocktails that induce maturation of the DCs, which are also pulsed with tumor antigens in the form of tumor lysate or peptides [70].

Many aspects of DC-based cancer vaccine production have several possible variations, such as the method of ex vivo DC generation, choice of DC subset, maturation and loading methods, and route of administration [68]. It has not been determined which DC subset would be best for immunotherapy purposes and it may vary depending on the situation. However, mo-DCs are used the most often for DC-based vaccines due to technical aspects, including their availably and ease of production [71]. Mo-DCs are considered an acceptable choice for the DC subset since they can powerfully activate tumor-specific CTLs and recruit other leukocytes to the local environment by secreting pro-inflammatory cytokines and chemokines [72]. However, the advent of enhanced methods for the isolation of naturally occurring DCs has led to a new generation of DC-based vaccines that consist of naturally occurring DCs rather than mo-DCs. There have been studies investigating the safety and efficacy of these vaccines, which have shown similar results to the mo-DC-based vaccines [1,73]. There is substantial debate over which DC subset would be best for the vaccine platform. There have not been any clinical trials comparing the efficacy of the different DC-based vaccine subsets [1]. However, the optimal DC subset for vaccination purposes may depend on specific cancer and patient circumstances, since each subset has different functional capacities [74].

DC-based vaccines were thought to hold a lot of promise in the field of cancer immunotherapy due to their desirable safety profiles and demonstrated immunogenicity. There was substantial excitement surrounding DC-based vaccines after the first one was approved by the United States Food and Drug Administration for use against advanced prostate cancers in 2010 [75]. Unfortunately, clinical trial results have been discouraging with overall response rates observed between 8 and 15 percent [1,2], which has left some investigators skeptical about the potential of DC-based vaccines. Therefore, there is room for improvement and optimization of DC-based vaccination [68,76].

DC-based vaccines are a great concept that in practice appear to have fallen short of expectations. There have been several reasons speculated for the limited clinical efficacy observed and many are related to the need for optimization of vaccine protocols [77]. Many aspects of DC-based vaccine generation and administration are varied throughout clinical trials and are a subject of debate in scientific literature, which can be attributed to limitations in the knowledge of DC biology and the DC-based vaccine immunobiology [2,78]. Therefore, further investigation into DC-based vaccine crosstalk with other leukocytes could provide insight and assist in the development of the next generation of more effective DC-based cancer vaccines.

The current clinical trials involving DC-based cancer vaccines have been reviewed [70], giving a comprehensive overview of 25 publications and 55 ongoing studies in various cancers, including glioblastomas, melanoma, and pancreatic cancers. The study covers the clinical trials that used either DC vaccines as a single agent or combined with other treatments such as chemotherapy. Other studies [79] also examined the current DC vaccines and the progress that has been made in the research. The platform of DC-based vaccines has many variabilities in its design, for instance, the subset of DC used, the method of vaccination preparation, the route of administration, the co-administration of DC vaccines with other therapies, and the obstacles facing efficacious DC vaccination. The publication mentioned above provided a summary of the ongoing clinical trials involving DC vaccines and outlined the type and stage of cancers, the clinical trial phases, the DC vaccine platforms used, and the main outcomes of the clinical trials. Most of the clinical trials were either phase I or II trials and utilized autologous DCs. A detailed description of the different types of DC vaccines, the challenges, and advanced observations in the current literature and clinical trials have been reviewed by another study [80]. Detailed analyses of the current and future state of DC vaccines in human clinical trials are also discussed and referred to in the current literature [70,79,80].

As the technology in medicine and bioinformatics continues to develop and improve, there is a parallel advancement in personalized medicine. The improved identification of neoantigens to promote better anti-tumor responses when made the target of a tumor vaccine has made tumor neoantigen vaccines an active area of research. There are a few different tumor neoantigen vaccines, including DC-based, nucleic acid, long synthetic peptide, and tumor cell vaccines. The advancements in this field of neoantigen vaccines are reviewed thoroughly by Peng et al., 2019 [81] and the various vaccines for cancer and their current state in research are briefly described by Schiller et al., 2022 [82].

## 4. Potential Role of Neutrophils in DC Vaccination

The observed interactions and crosstalk between DCs and neutrophils suggest neutrophils may be playing roles in the immunobiology behind DC-based vaccines. We have previously demonstrated potential roles that neutrophils are playing in the context of another immunotherapeutic platform, oncolytic vesicular stomatitis virus, in mouse models [83]. Furthermore, we have observed communications between DC-based vaccines and neutrophils that have implications for the immune responses elicited by the vaccine (unpublished data). We believe there are communications between DC vaccines and neutrophils that are not fully understood and the roles of neutrophils and their heterogeneity in the immunobiology behind DC vaccines are underappreciated. The crosstalk occurring between neutrophils could have implications for the design of future generations of DC vaccines. To our understanding, the degree to which neutrophils contribute to the immune responses elicited by DC vaccines is unknown since there is limited research into the exact roles neutrophils play in vaccine immunobiology. To take the relationship between neutrophils and DCs into account when strategizing and designing therapeutics, we must first have a comprehensive understanding of the interplay between neutrophils and DCs that already exists in DC vaccine immunobiology so that we can best decide how to therapeutically manipulate them. Figure 1 illustrates some of the mechanisms in which neutrophils have been observed to communicate with DCs and/or influence adaptive immune responses.

Neutrophils have been shown to contribute to the formation of T cell responses through direct antigen presentation [11,12] as well as being a source of antigens for DCs, which then educate T cells [17]. Therefore, neutrophils could be playing roles in the education of adaptive immunity in cancers and during immunotherapies such as those that involve DCs. It was demonstrated that DC-neutrophil interactions were important in vaccine immunity induced against live attenuated Leishmania [84], highlighting the significance DC-neutrophil crosstalk can have in a vaccine platform. In this instance, DCs were observed to phagocytose neutrophils infected with the parasite and had improved T cell priming in the draining lymph nodes compared to when neutrophils were depleted. In the case of DC-based vaccinations, DCs may already have antigens loaded and have undergone priming. However, as mentioned previously, DC vaccines have several limitations and inadequacies, including the delivery/migration of the DC vaccine to the lymph nodes for education of adaptive lymphocytes. If the DC vaccine has restricted infiltration/migration within the host or the survival of the DCs in the vaccine is limited due to non-optimized routes of administration, dosing, or preparation of the DCs themselves, neutrophils could have the potential to pick up the antigens loaded on the DCs and bring them to endogenous DCs to prompt T cell responses. Furthermore, neutrophils may also help with assisting to prime naïve host DCs. If the DC vaccine has primed tumor antigen-specific responses and cytotoxic effector cells have begun targeting and killing tumor cells, this can result in the release of more tumor antigens that neutrophils could pick up and deliver to endogenous DCs for education. Moreover, the ability of neutrophils to promote the recruitment and activation of DCs [16,85] could also be assisting in these circumstances. Therefore, neutrophil and endogenous host DC crosstalk could assist in priming and augmenting the anti-tumor antigen-specific T cell repertoire during already ongoing immune responses elicited by treatment with exogenous DC vaccination and could be useful if the tumor were to undergo immune editing.

Interestingly, the crosstalk between neutrophils and DCs during the inoculation with the live attenuated Leishmania vaccine resulted in increased DC responses, whereas during the infection with the wildtype parasite, the neutrophil-DC interaction was observed to reduce DC activation [84]. Therefore, the interactions between neutrophils and DCs appears to be incredibly complex, which stresses the need to study and gain a better understanding of their crosstalk and how various circumstances can influence it.

As mentioned, neutrophil production of NETs has been associated with pro-tumorigenic immune responses. Regarding NETs in DC-neutrophil interactions, NETs have been implicated in promoting activation of DCs in the early stages of inflammation. However, extended exposure of NETs can result in DCs undergoing cell death [86]. Consequently, this neutrophil-DC interaction may hinder the formation of adaptive immunity and limit inflammatory immune responses necessary to combat disease. NETs have also been shown to induce production of chemokines that attract neutrophils, indicating that the production of NETs can result in amplification of neutrophil responses that might further contribute to pro-tumorigenic immunity. Conversely, DCs have been shown to be able to degrade NETs through the secretion of DNase1L3 [87]. This indicates that in tumor immunity, DCs could be attenuating pro-tumoral neutrophils by degrading NETs and hindering NET-induced neutrophil recruitment. On the other hand, NETs may be hindering DC responses and suppressing anti-tumoral immunity. Figure 2 shows some of the ways neutrophils could be playing roles in DC vaccine immunobiology. However, this DC-neutrophil crosstalk will need to be studied within a cancer context to gain a better understanding of these interactions and to verify if these interactions extend to the unique immunological settings of cancer-bearing hosts.

Since neutrophils are first responders and can release NETs when stressed, the potential roles of NETs in post-surgical recurrence of tumors were investigated. Following surgery in patients with metastatic colorectal cancer, there was an association between patients with increased NET formation and poorer prognoses. This was examined further in models of surgical stress and colorectal cancer in mice. In the murine models, it was demonstrated that following surgical stress, there was an increase in metastasis, which was reduced when the mice were either given DNAse or had the enzyme peptidylarginine deaminase inhibited, both of which act to inhibit NET formation [45]. Further evidence has demonstrated that NETs may be playing roles in facilitating resistance to cancer therapies. For instance, NETs were studied in a murine model of bladder cancer and mice were found to have more NETs in their tumors following radiation and that removing/blocking NET formation improved responses to radiation therapy. This NET-associated resistance mechanism appeared to involve limiting CD8+ T cell responses. Data from patients with muscle-invasive bladder cancers supported these findings [88]. This research supports the use of therapies targeting the production of NETs as adjuvant therapy.

NETs have been shown to shield tumor cells from cytotoxic leukocytes, including NK cells and CD8+ T cells, resulting in impaired clearance of tumors [40]. We have shown that DC vaccines elicited anti-tumor efficacy by engaging both NK and CD8+ T cells [89]. Additionally, in murine models and in vitro experiments using samples from patients with metastatic colorectal carcinomas, NETs were shown to have the capacity to induce suppression of T cell function through promoting T cell exhaustion [90]. Therefore, the formation of NETs may be hindering DC vaccine efficacy by interfering with DC-induced NK and T cell anti-tumor responses. This implies that enhanced anti-tumor responses might be obtained through combining DC vaccines and NET inhibitors.

The inhibition of NET formation has become an intriguing therapeutic target, and several pre-clinical studies evaluate the potential to inhibit or block NET production [91]. However, when blocking or inhibiting NET formation, several potential consequences also need to be considered, for instance, the loss in pathogenic protection associated with the function of NETs. Multiple agents can block or inhibit the production of NETs; therefore, several factors should be examined for which would work best with DC vaccine therapies. The use of anti-NET therapies in conjunction with immune checkpoint blockade has recently been considered an attractive therapeutic strategy [92]. Indeed, the use of therapies to target NET formation parallels immunotherapies to enhance anti-tumor immunity.

In the TME, neutrophils can adopt a pro-tumorigenic or anti-tumor phenotype that can be dictated by external stimuli that they receive [21,30,31]. Pro-tumorigenic neutrophils play significant roles in the progression and development of advanced cancers [27,28,29,50,57,60]. As a result, they have gained considerable attention in the field of cancer biology, and therapeutics that target neutrophil phenotypes and plasticity in TMEs is an active area of research [50,58,60]. Since DC cancer vaccines function to alter the immunosuppressive state of TMEs to produce an immunogenic environment and can be manipulated to express specific phenotypes and cytokine profiles [68,77], they represent a possible immunotherapeutic approach for influencing neutrophil phenotypes and functions. The plasticity between N1s and N2s and how external stimuli can influence them is an active area of research [33,58,59] and should be studied in the context of DC vaccines to evaluate the degree to which DC vaccines currently influence TAN polarity and if DC vaccines could be strategically prepared to promote N1 phenotypes, for instance via promotion of production of specific cytokine or enzyme combinations.

## 5. Conclusions

Neutrophils are critical players in cancer pathogenesis and anti-tumor immunity. The plasticity in neutrophil functionality and the increased neutrophil ratio observed in cancers implicate neutrophils as an attractive therapeutic target whereby their phenotypes can be manipulated to promote desired responses in the TME. Moreover, NETs have been implicated in contributing to resistance to radiation [88], immunotherapy [40,48], and post-surgery complications [45]. NETs have also been shown to have roles in supporting T cell exhaustion [90]. Therefore, therapeutic interventions targeting the formation of NETs in combination with other therapies is an attractive area of research.

Although DC vaccines are a promising immunotherapeutic tool for cancers, there is still a need to optimize this platform, which could be achieved by exploiting their crosstalk with other leukocytes. DC vaccines are amenable to manipulations that could influence the immune system and the TME, which could be strategized toward targeting host neutrophil functionality as well. Furthermore, the efficacy of DC vaccines could be enhanced with appropriate combination therapy. DC vaccines, combined with a targeted neutrophil treatment, could be an exciting avenue of research that could potentially have synergistic effects by simultaneously augmenting the functions of both cell types and promoting positive crosstalk between them. For instance, NET formation appears to have a negative influence on TMEs, and NETs have been suggested to limit cytotoxic effectors, NK, and CD8+ T cells [40], which are the main cells that DC vaccines are targeted towards promoting the activity of. Therefore, combining inhibitors of NET formation and DC vaccines could enhance the anti-tumor responses in the TME elicited by the DC vaccine and hinder the pro-cancer responses associated with NET formation, such as metastasis. This should be investigated preclinically using samples from patients undergoing treatment with the DC vaccine in ex vivo experiments evaluating what is currently happening with neutrophil responses during DC-based vaccinations. Furthermore, additional preclinical studies should be conducted in vivo in animal models to allow for greater manipulation and investigation of anti-NET therapeutics in combination with DC vaccination treatments. These findings should be applied to findings from human patient samples.

Essentially, to improve DC vaccine strategies, there is a need for a better understanding of the immunobiology occurring behind the vaccine; for instance, their communications with another key player in cancer that has been previously overlooked, neutrophils.

## Figures and Tables

**Figure 1 ijms-24-00896-f001:**
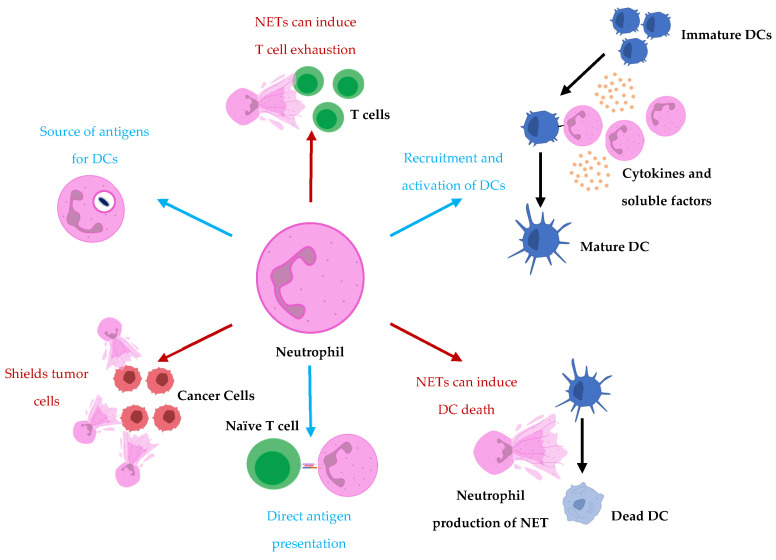
Neutrophils influence DC vaccine efficacy either to support the formation of or suppress adaptive immune responses. Neutrophils recruit and activate DCs and act as either a source for antigen uptake by DCs or a direct presentation of antigens to T cells. Conversely, neutrophils have also been shown to hinder T cell responses by supporting T cell exhaustion. Furthermore, NETs have been implicated in tumor cell protection from cytotoxic effector cells by acting as shields. Moreover, prolonged exposure of DCs to NETs has been shown to promote cell death of DCs.

**Figure 2 ijms-24-00896-f002:**
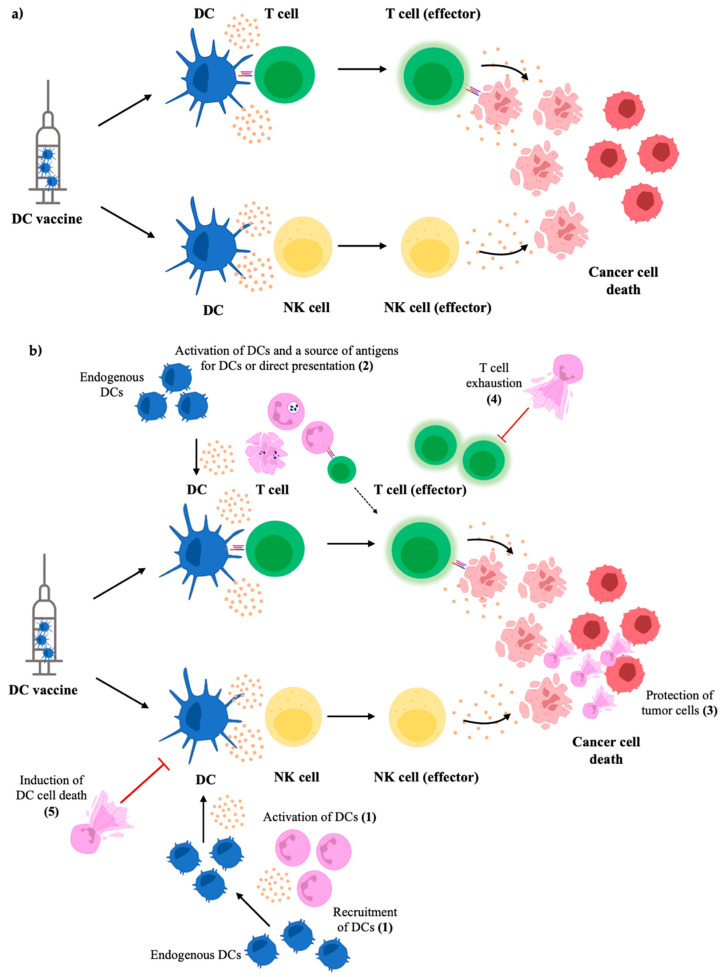
Possible mechanisms of how neutrophils may influence DC vaccine immunobiology. The general concept of DC vaccines is to administer primed and mature DCs to the host for presenting tumor-associated antigens, stimulating effector NK cells, and activating antigen-specific cytotoxic T cells to kill tumor cells (**a**). The influence of neutrophils on these responses has not been thoroughly investigated; however, known crosstalk and interactions suggest they could play active roles in the vaccine’s immunobiology. Neutrophils have been observed to recruit and activate DCs, enhancing the endogenous DC responses to propagate more robust anti-tumor responses (**1**). Additionally, the anti-tumor responses induced by the DC vaccine can result in neutrophils taking up tumor antigens and either directly presenting them to T cells or acting as a source of antigens for endogenous DCs (**2**). However, NETs could hinder DC vaccine efficacy by shielding and protecting tumor cells from cytotoxic effector cells primed by the vaccine (**3**) and inducing T cell exhaustion (**4**), thereby suppressing tumor antigen-specific responses. Extended exposure to NETs has also been shown to cause cell death of DCs (**5**), which could limit anti-tumor responses (**b**).

## Data Availability

Not applicable.

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
