# Peer review of "Neutrophils in Dendritic Cell-Based Cancer Vaccination: The Potential Roles of Neutrophil Extracellular Trap Formation"

_ijms, 2023, doi:10.3390/ijms24020896_

Round 1

Reviewer 1 Report

Neutrophils have seemingly conflicting roles in the process of cancer initiation and progression, as well as in the process of cancer metastasis and relapse, likely due to the multifaceted roles of neutrophils in tumor microenvironment and inflammation.  DC cells are important antigen presenting cells, and their application as vaccine carrier for cancer treatment has been implored but with a limited patient response. This review summaried the role of neutrophil extracellular traps in cancers and discussed the potential to combine DC vaccine and NET limitation for the treatment of cancer. This area is of potential for the interest of a reader audience in the area of cancer vaccine biology, neutrophils, dendritic cells, and drug development.

Line 42: “Neutrophils are another innate leukocyte in the immune system. ” Consider rewriting this sentence.

Line 53-54: 

The life span of neutrophils has a wider range depending on various studies, please reconsider the statement made in lines 53-54.

In the Subtitle 3 - DC vaccines section, 

a.     Please expand on what types of targets that DC vaccines have been designed for.  

b.     Please compare the DC vaccine cell therapy approach with the other type of vaccine treatment for tumors, especially in the context of tumor neoantigen vaccines.

Figure 2:

Figure 2a is a part of figure 2b, it can be eliminated.  Please label with numbers/letters each of the mechanisms underlying the regulation of DCs and cancers by neutrophils in the current figure 2b and refer to these mechanisms by labeling in the figure legends.

Informed consent statement:

Does the review involve human subject?  This part needs to be revised.

Author Response

"Please see the attachment".

Reviewer 2 Report

This is an intriguing review article demonstrating the potential of using DC-based vaccine and modulation of NET for cancer therapy. The rationales and academic clues in this manuscript are sound. The concepts are well presented and the logics are clear. There are some minor issues to be clarified.

1.  Since NETosis inhibitors might be able to enrich the anti-tumor efficacy of DC-based vaccine, a comprehensive and updated review paper on anti-NET for disease treatment (Molecular and Cellular Biochemistry, 2022, 477:673–88) should be cited and briefly discussed.

2.  The phase-I or II human clinical trials on DC-based vaccines should be reviewed and described.

3.  The optimal preclinical models that may validate the DC-based vaccine/anti-NET therapeutic strategy should be discussed.

4. Please check if the reference format is correct for IJMS. 
